Mitigating inappropriate concepts in text-to-image generation with attention-guided Image editing

Oh Jiyeon
Jeong Jae-Yeop
Hong Yeong-Gi
Jeong Jin-Woo jinw.jeong@seoultech.ac.kr
Department of Data Science, Seoul National University of Science and Technology , Seoul , Republic of South Korea
Comai Sara
Electronic publication date: 2025 Sep 9
Publication date: 2025
Volume: 11
Electronic Location ID: e3170
Received 2024 Oct 17; Accepted 2025 Aug 7
Copyright: © 2025 Oh et al.
Copyright year: 2025
Copyright holder: Oh et al.
License: This is an open access article distributed under the terms of the Creative Commons Attribution License, which permits unrestricted use, distribution, reproduction and adaptation in any medium and for any purpose provided that it is properly attributed. For attribution, the original author(s), title, publication source (PeerJ Computer Science) and either DOI or URL of the article must be cited.
License URL: https://creativecommons.org/licenses/by/4.0/

Keywords: Text-to-image generation, Inappropriateness mitigation, Attention map, Deep learning

Funding: Research Program funded by the SeoulTech (Seoul National University of Science and Technology) This study was supported by the Research Program funded by the SeoulTech (Seoul National University of Science and Technology). The funders had no role in study design, data collection and analysis, decision to publish, or preparation of the manuscript.

==============================
Text-to-image generative models have recently garnered a significant surge due to their ability to produce diverse images based on given text prompts. However, concerns regarding the occasional generation of inappropriate, offensive, or explicit content have arisen. To address this, we propose a simple yet effective method that leverages attention map to selectively suppress inappropriate concepts during image generation. Unlike existing approaches that often sacrifice original image context or demand substantial computational overhead, our method preserves image integrity without requiring additional model training or extensive engineering effort. To evaluate our method, we conducted comprehensive quantitative assessments on inappropriateness reduction, text fidelity, image consistency, and computational cost, alongside an online human perceptual study involving 20 participants. The results from our statistical analysis demonstrated that our method effectively removes inappropriate content while preserving the integrity of the original images with high computational efficiency.

Introduction

Various online text-to-image generation tools based on diffusion, such as DALL-E (OpenAI, 2024) and Stable Diffusion (SD) (Rombach et al., 2022), have become widely accessible for artistic and entertainment purposes. These services have gained immense popularity due to their ability to produce high-quality images with remarkable efficiency, leading to an unprecedented volume of artificial intelligance (AI)-generated visual content. In the AI-assisted image generation process, end-users typically engage in an iterative process, refining their prompts based on initial outputs to achieve their desired results. However, image generation models can unintentionally incorporate undesirable concepts learned from large-scale, unrefined training data. Therefore, generated images may contain elements of racism, copyright infringement, or other problematic content, potentially eliciting negative reactions from users who find such content offensive or distressing. These issues have raised significant social concerns (Bird, Ungless & Kasirzadeh, 2023), including those related to copyright and privacy infringement (Eloundou et al., 2023; Carlini et al., 2023; Franceschelli & Musolesi, 2022), as well as biases related to disability (Bianchi et al., 2023) and religion (Bird, Ungless & Kasirzadeh, 2023). To mitigate this issue, users often resort to repeated prompt refinement in an attempt to induce the generation of appropriate images. However, this approach is fraught with limitations. There is no guarantee that the recreated image will maintain the desired style of the original image the users wanted to use while successfully eliminating all inappropriate content. Moreover, this process of repetitive manual correction is not only tedious but can significantly detract from the user experience (Shneiderman et al., 2016).

Consequently, it is a significant challenge to support the prevention of inappropriate image generation while maintaining their original style and quality. One widely adopted solution is to censor the training data set in order to prevent generative models from learning inappropriate representation (Rando et al., 2022; Rombach et al., 2022). While conceptually straightforward, this approach is labor-intensive and lacks adaptability, as incorporating new data necessitates repeated censoring and training. For example, even though Rombach et al. (2022) trained SD models using LAION-5B (Schuhmann et al., 2022), which had inappropriate images explicitly removed, SD still occasionally produces inappropriate content. Additionally, a significant drawback of censoring approaches is the potential deterioration of output quality due to reduced dataset size (Gandikota et al., 2023). Alternative methods using textual cues to guide the generative process and mitigate inappropriate outputs, such as Safe Latent Diffusion (SLD) and Erase Stable Diffusion (ESD), have been presented Schramowski et al. (2023) and Gandikota et al. (2023), respectively. Specifically, SLD employed post-hoc prevention by adjusting network parameters to avoid generating problematic outputs without additional training. However, the SLD often generates images that significantly deviate from the original, potentially compromising the user’s intended artistic vision and style. Moreover, it involves manual adjustments of numerous hyper-parameters, complicating the inference process to generate optimal image outputs. Conversely, ESD addressed inappropriateness reduction while maintaining the original image style, but requires additional model training and adaptation, which can be inefficient and time-consuming.

To address these limitations, this study proposes an approach that utilizes attention maps of SD to mitigate the inappropriateness of generated images. We detected the inappropriateness presented in attention maps and reduce its representation, thereby guiding generative models toward creating images with reduced inappropriate aspects during the generation process. Notably, unlike previous methods, it does not require data filtering (Rando et al., 2022; Rombach et al., 2022), additional model training (Gandikota et al., 2023), or complex hyper-parameter adjustment (Schramowski et al., 2023), offering a straightforward yet effective solution.

Related work

Reducing inappropriate concepts in images

Inappropriate images from generative models are identified as a new social issue (Bird, Ungless & Kasirzadeh, 2023). Advanced diffusion models have demonstrated the ability to learn and reproduce undesirable concepts, largely due to their training on extensive internet-sourced datasets. These datasets, often compiled using search engine criteria, may include personal, offensive, and hateful imagery (Wu et al., 2023; Li et al., 2024). Early attempts to solve this problem have been relatively straightforward yet limited in their effectiveness. For example, a user can re-generate images with their own prompt editing until an image without inappropriate content is provided. However, this approach is time-consuming, user/skill-dependent (e.g., level of prompt engineering), and significantly alters the original creative vision or purpose of the image generation. Data filtering is another obvious solution that removes inappropriate content from training data. However, this approach is also time-intensive and may compromise output quality by reducing the size of data (Gandikota et al., 2023). Post-generation techniques have emerged as alternative strategies. These methods have focused on modifying or obscuring inappropriate content after image creation, employing strategies like content masking (Maidhof et al., 2022) and targeted image editing (e.g., removing nudity by putting on clothes) (More et al., 2018). However, these approaches still struggled with producing natural outputs and required additional model training or image processing steps.

Recent research has explored leveraging text data to identify potentially inappropriate content and refining the image generation process. One notable method is SLD (Schramowski et al., 2023), which extends the capabilities of SD by modifying the generated guidance for text using classifier-free guidance to reduce the inappropriateness of images. During inference, SLD employs a set of hyper-parameters to guide SD in the direction of generating appropriate images. However, empirical observations revealed a significant trade-off: stronger content regulation often results in output images that deviate considerably from the original. Furthermore, considerable effort was required to adjust multiple hyper-parameters. Another approach called ESD was proposed by Gandikota et al. (2023), which aims to erase unsafe concepts through minimal training procedures. ESD employs a fine-tuning process to remove specific undesirable concepts from the weights of the pre-trained SD model. To this end, they utilized a teacher model trained with negative prompts to guide pre-trained SD in eliminating visually unsafe concepts. ESD demonstrated performance as effective as SLD (Schramowski et al., 2023), despite relying primarily on fine-tuning rather than extensive retraining.

While SLD and ESD have made significant strides in addressing inappropriate content generation, each approach comes with its own set of trade-offs. SLD offers adaptable content moderation but demands precise parameter tuning and often significantly alters the original styles. On the other hand, ESD enables specific concept removal but necessitates additional model training. Our study aims to streamline the process of reducing inappropriate content while enhancing the image-generation experience for users. Unlike ESD, our method eliminates the need for additional training, and in contrast to SLD, it retains the style of original images and reduces the effort to identify optimal hyper-parameters. To achieve this, we propose to leverage the attention maps of diffusion models.

Image editing using attention maps

Recent research has seen a proliferation of techniques leveraging attention maps for image and video editing (Liu et al., 2024b, 2024a; Hertz et al., 2022; Chefer et al., 2023), offer efficient, tuning-free or minimal-training approaches. These methods leverage the rich spatial and semantic information encoded within attention mechanisms to enable targeted modifications. Specifically, text-to-image diffusion models establish a spatial correspondence between individual words in the input prompt and specific regions in the generated image. This correspondence manifests through distinct attention maps for each word, indicating that the semantic information associated with each term can be localized to particular areas within the visual output. Accordingly, altering the prompt leads to corresponding changes in the attention maps. For example, Hertz et al. (2022) explored revising images while preserving original styles by manipulating cross-attention maps between text and spatial layouts. Another method, MasaCtrl (Cao et al., 2023), was introduced to address the challenges of complex non-rigid image editing. This method transforms the self-attention in diffusion models into cross-attention, thereby facilitating access to the images’ feature representations, encompassing both local content and textural elements. Consequently, this enabled sophisticated editing while maintaining image coherence. Furthermore, Chefer et al. (2023) demonstrated object-specific editing in SD by exploiting cross-attention maps. In the case of video editing, Liu et al. (2024b) presented Video-P2P, a large-scale model that employed separate unconditional embedding for both source and target prompts, thereby enhancing both reconstruction fidelity and editability. Their works revealed that utilizing attention map variations allows for edits that effectively preserve the structure and content of the original image, enabling fine-grained control over the generated output through strategically altered text input.

Despite these advancements in leveraging attention maps for image and video editing, their potential for inappropriate content mitigation in generative models remain largely unexplored. Existing solutions, such as ESD and SLD, often either compromise original image style or demand substantial computational overhead for model fine-tuning and complex hyperparameter tuning, which severely limits their practical usability. To address this gap, our work investigates the feasibility of using cross-attention maps in diffusion models. By strategically leveraging attention mechanisms, specifically by harnessing both classifier-free guidance and directly editing attention maps, we aim to filter out the representation of inappropriate concepts in generated images while preserving their intended visual characteristics. By exclusively utilizing pretrained text-to-image models and minimizing hyperparameter complexity, this approach allows us to suppress undesirable content without the need for additional model training or extensive engineering effort. To thoroughly evaluate our attention map-based content mitigation framework, we pose the following research questions: RQ1 Can our attention map-based approach effectively and efficiently mitigate inappropriate content?

RQ2 How do humans perceive the appropriateness and quality of the images generated by our attention map-based approach?

RQ3 What is the impact of key hyper-parameters on the performance of our attention map-based approach?

Method

In this section, we presented our approach to reducing inappropriate content in images generated by latent Diffusion models. First, we described the background of our method, including the use of classifier-free guidance and cross-attention mechanisms. Then, we discussed the detailed process of attention-guided image editing for mitigating inappropriate content.

Background

Latent diffusion models (LDMs) (Rombach et al., 2022) generate an image latent z0 using a random noise vector zT and textual condition p as inputs. They predict and remove artificial noise εt added to zt over T steps, resulting in z0, which is decoded to generate the image. To facilitate this iterative denoising process, the model predicts the noise εθ to refine the latent zt−1 from zt using the following equation:

(1) zt−1=1αt(zt−1−αt1−α¯tεθ(zt,t,P))+σtε,

where αt¯ is the cumulative product of denoising strength α up to timestep t, P represents the embedding of the text input p, and σt is the standard deviation of the added noise.

Classifier-free guidance

To mitigate the amplification effect of text conditioning during the inference process, classifier-free guidance was proposed (Ho & Salimans, 2022), enabling the model to interpolate between the conditional and unconditional noise predictions:

(2) ε~θ(zt,t,P,∅)=g⋅εθ(zt,t,P)+(1−g)⋅εθ(zt,t,∅),

where ∅ denotes the embedding of a null text, and g is the guidance weight. Classifier-free guidance aims to balance the influence of textual conditioning to enhance the stability and semantic alignment of the generated images.

Cross-attention in stable diffusion

Building upon the principles of LDMs, SD further refines the image generation process. One of the key enhancements is the integration of cross-attention layers within a U-Net architecture (Ronneberger, Fischer & Brox, 2015), which allows for more precise alignment between the textual input and the generated image features. The cross-attention mechanism operates by integrating textual information directly into the image generation process using query (Q), key (K), and value (V) matrices. Specifically, Q is derived from image features zt through learned linear transformation, while K and V are projected from the textual embedding P using learned linear transformations. The cross-attention map is defined as:

(3) Q=WQzt,K=WKP,V=WVP

(4) Mcross=Softmax(Q(K)Tdk),

where WQ, WK, and WV are learned weight matrices, and dk is the dimension of key vectors. The cross-attention map enables the model to highlight specific regions of an image that are closely associated with textual features during the generation process, thereby enhancing the fidelity and semantic coherence of visual representations to textual descriptions.

Attention-guided image editing for mitigating inappropriateness

Building on the insight that prompt alterations correspond to changes in attention maps (Hertz et al., 2022; Chefer et al., 2023), we propose an effective approach that regulates these maps to mitigate inappropriate content in generated images. Our approach is based on the assumption that if a text prompt contains inappropriate words/terms, then the corresponding attention maps are likely to highlight regions associated with inappropriate elements. Therefore, we propose to use attention maps derived from inappropriate words to identify and mitigate the presence of inappropriate elements during image generation.

Figure 1 shows an illustrative example of how we manipulated attention maps to reduce inappropriateness in the image generation process. Let Mtext denote the attention map derived from a text prompt embedding Ptext, and Minappr denote the attention map derived from the embedding of inappropriate words Pinappr. Given an image I generated from the predicted latent z0, Mtext illustrates how Ptext influences each part of I. Conversely, Minappr highlights areas within I where inappropriate content is prominent, as indicated by higher attention values. Therefore, it can be interpreted that the residual attention map Mr, derived from the difference between Minappr and Mtext, specifically identifies image regions containing inappropriate elements. Thus, by utilizing Mr, which captures areas of higher inappropriateness, one can effectively create an image where inappropriate content has been mitigated.

Figure 1 Overview of our attention-guided inappropriate concept mitigation framework with example attention maps (class: sexual).

Darker regions in the map indicate higher attention values. The “Original” image is generated by stable diffusion using only the text prompt, without applying our mitigation approach. Note that the displayed attention maps are manually constructed for illustrative purposes to convey the conceptual mechanism of our approach. All sensitive contents have been intentionally blurred.

The pseudo algorithm is shown in Algorithm 1. The core of the algorithm is an iterative denoising process that runs from timestep T to 1. The proposed approach leverages classifier-free guidance to generate high-quality images that align with the given prompts while mitigating inappropriate content. Therefore, for the initial input, we utilize unconditional embedding ∅ along with textual Ptext and inappropriate word Pinappr embeddings. The algorithm starts with a randomized latent vector zT sampled from a standard normal distribution. For the first few steps (determined by τ), the algorithm uses standard diffusion with only the text prompt, which allows the initial structure of the image to form based on the user’s intention (Line 13). After these steps, we apply a mechanism for inappropriateness reduction through Line 5–11. For this, we get attention maps for both the text prompt Mtext and the inappropriate words Minappr (Line 7). To reduce the inappropriateness within Mtext, we first compute a residual attention map Mr by subtracting the text attention from the inappropriateness attention (Line 8). Then, we apply ReLU activation to Mr (Line 9), which suppresses negative values, emphasizing areas where the influence of inappropriate content exceeds that of text prompts. Subsequently, a final attention map Mt is created by subtracting the residual map Mr scaled by λ from the text attention map Mtext. Here, λ modulates the extent of adjustment applied to Mtext. The process continues until the final denoised latent z0 is obtained, which is then used to generate the final image I.

Algorithm 1 Mitigating inappropriate concepts in image generation.

1: Input: text prompt embedding Ptext, inappropriate words embedding Pinappr, unconditional embedding ∅	
2: Output: Image without inappropriateness I	
3: Initial randomized latent zT∼N(0,1)	
4: for t←T,…,1 do	
5:  if t<T−τ then	
6:    zt−1← DM( zt,Ptext,Pinappr,∅,t) {	
7:    Mtext, Minappr←attε(Ptext,Pinappr,∅)	
8:     Mr←Minappr−Mtext	
9:     Mr←ReLU(Mr)	
10:      Mt←Mtext−λMr	
11:    }	
12:  else	
13:    zt−1← DM( zt,Ptext,∅,t)	
14:  end if	
15: end for	
16: return I←Generate(z0)	

The proposed method allows for real-time filtering of inappropriate content during the generation process without requiring model retraining or extensive hyper-parameter tuning. It can also strike a balance between appropriateness and preserving the user’s original artistic intent.

Evaluation

To evaluate the performance of our proposed model, we conducted both quantitative and qualitative assessments as well as a human perceptual study. In our quantitative evaluation, we validate the effectiveness of the proposed approach in (1) reducing inappropriate content and (2) preserving the original semantics in generated images. Additionally, we examine the model’s ability to reflect the context of text prompts in generating images. In the qualitative evaluation, we provide a series of illustrative examples that can demonstrate the comparative quality of images generated by our model and the baseline approaches. Finally, we analyze how humans perceive the appropriateness, quality, and overall impressions of the generated images through the result of our human perceptual study.

Experimental settings

Dataset

In this article, we utilize the Inappropriate Image Prompts (I2P) evaluation dataset (Schramowski et al., 2023), which is designed as a standardized benchmark for evaluating the propensity of text-to-image models to generate inappropriate content. The I2P dataset consists of seven classes of images: “illegal activity”, “hate”, “self-harm”, “violence”, “shocking”, “sexual”, and “harassment”. The dataset includes 4,703 unique text prompts, retrieved from Lexica website (https://lexica.art). Each prompt may be assigned to at least one of the seven classes. In addition to these prompts, it provides essential hyper-parameters such as seed, guidance scale, and image dimensions for re-productibility in generating images using SD. We utilized the entirety of I2P dataset as a test set to assess the safety performance of the evaluated models.

Evaluation metrics

To evaluate the effectiveness in reducing inappropriate content, we follow the I2P test bed established by Schramowski et al. (2023), which integrates Q16 (Schramowski, Tauchmann & Kersting, 2022) and NudeNet (notAi tech, 2019) classifiers. In the I2P test protocol, an image is classified as inappropriate if either classifier detects the presence of inappropriate content based on its respective labels. The final score of the I2P evaluation is calculated based on the ratio of inappropriate images detected to the total number of generated images per class.

Furthermore, to evaluate how well the generated images preserve the original context, we employ learned perceptual image patch similarity (LPIPS) (Zhang et al., 2018). This measure quantifies the visual difference between the original images generated by SD and those generated using inappropriate content reduction methods.

Finally, we evaluated how accurately the generated images reflect the context of the input text prompt. For this, we utilize CLIP score (Radford et al., 2021) to measure the semantic similarity between input text prompts and corresponding generated images.

Baseline

We compared the performance of our method against SLD (Schramowski et al., 2023) and ESD (Gandikota et al., 2023) methods. Specifically, in the case of SLD, we used the ‘Weak’ (SLD-Weak, hereafter) and ‘Max’ (SLD-Max, hereafter) versions. The SLD-Max version, with increasing aggressiveness of changes on the resulting image, is known to demonstrate superior efficacy in eliminating inappropriate content compared to the SLD-Weak. For the ESD model, we utilized the ESD-u-1 version. This model, similar to the proposed method, also prioritizes maintaining high similarity to the original images, while effectively reducing inappropriate content in the generated images.

Data analysis

To rigorously evaluate the performance of our proposed method and to statistically validate the observed differences among comparisons, we conducted a series of statistical tests across all quantitative evaluations. For I2P evaluation, which yields a binary outcome, we used Cochran’s Q Test to determine the overall statistical significance across conditions. If an overall difference was detected, pairwise McNemar tests were subsequently applied to identify specific pairs of conditions that showed significant differences. The Cohen’s g was employed as the effect size measure for these comparisons. Following Cohen’s guideline, g values of 0.15, 0.25, and above 0.25 are interpreted as small, medium, and large effects, respectively. For both LPIPS and CLIP scores, which are continuous measures, we employed the Friedman test to assess overall differences across conditions. Following a significant Friedman test, pairwise Wilcoxon Signed-Rank tests were performed to identify specific significant pairs with the Rank-biserial correlation ( r) reported as the effect size. As a conventional interpretation guideline, r values of 0.1, 0.3, 0.5, and above 0.5 represent negligible, small, medium, and large effects. For all pairwise comparisons, p-values were adjusted using Bonferroni correction.

Implementation details

Our method is developed based on SD version 1.4. In our experiments, we configured the hyper-parameters within Algorithm 1; the editing initiation τ to 0 and attention map scaling ratio λ to 0.3, with 50 denoising steps (T) for attention map controls. For inappropriate words, we utilized the list of class labels from the I2P dataset. All generated images are of size 512 × 512, consistent across all methods, and all experiments are conducted on a single GeForce RTX3090 24 GB GPU.

Quantitative results

Inappropriateness reduction

Table 1 shows the I2P evaluation results, including scores for the seven classes and the overall score. To statistically validate these observations, we conducted a Cochran’s Q test on the overall I2P scores, which revealed a statistically significant difference among the methods ( Q=1,521.16, p<0.001). Subsequent pairwise McNemar tests revealed statistically significant differences between all pairs of conditions, as summarized in the first column of Table 2. Specifically, our method (0.33) differed significantly from ESD (0.31; p=0.029, g=0.084) and SD (0.36; p=0.002, g=0.112), both with small effect sizes. Notably, comparisons with SLD-Weak (0.25) and SLD-Max (0.09) resulted in highly significant differences ( p<0.001), with large effect sizes of g=0.283 and g=0.781, respectively. All other comparisons among the baseline methods also showed highly significant differences ( p<0.001) with medium to large effect sizes. Among the methods evaluated, SLD-Max demonstrated the highest performance, substantially reducing the probability of generating inappropriate content by over 75% (i.e., from 0.36 to 0.09 on the overall score). In contrast, ESD and our method reduced inappropriate content by 13% (0.36 to 0.31 overall) and 8% (0.36 to 0.33 overall), respectively. These differences in performance between SLD and ESD/Ours can be attributed to their distinct approach. SLD analyzes and eliminates all inappropriate elements from both text prompts and generated images. It achieved higher appropriateness through its inherent guidance mechanism that explicitly modifies the latent space, indirectly influencing the predicted noise to ensure only appropriate content is produced. However, the visual appearance of outputs from SLD-Weak and SLD-Max tended to be significantly different compared to the original image generated by SD, even though the same text prompt was used. Conversely, ESD and our method prioritize the preservation of the contexts of the original images, leading to more subtle visual alterations compared to the SLD approaches.

Table 1 I2P evaluation results ( ↓).

Shown are the probabilities of generated images classified as inappropriate.

Class	SD	SLD-weak	SLD-MAX	ESD-u-1	Ours	
Illegal activity	0.37	0.24	0.07	0.33	0.33	
Hate	0.39	0.25	0.10	0.29	0.34	
Self-harm	0.38	0.25	0.06	0.34	0.37	
Violence	0.41	0.32	0.15	0.39	0.38	
Shocking	0.50	0.39	0.15	0.39	0.48	
Sexual	0.22	0.11	0.04	0.15	0.19	
Harassment	0.31	0.24	0.08	0.29	0.27	
All	0.36	0.25	0.09	0.31	0.33	

Table 2 Pairwise comparison results for each quantitative evaluation.

Each cell reports the Bonferroni-corrected p-value and the corresponding effect size, with magnitude interpretation.

	Inappr. Reduction	Image similarity	Text-image alignment	
p-value	Effect size |g|	p-value	Effect size |r|	p-value	Effect size |r|	
SD vs. SLD-Weak	<0.001	0.372 (large)	–	–	0.811	0.033 (negligible)	
SD vs. SLD-Max	<0.001	0.803 (large)	–	–	<0.001	0.097 (negligible)	
SD vs. ESD	<0.001	0.189 (medium)	–	–	<0.001	0.069 (negligible)	
SD vs. Ours	0.002	0.112 (small)	–	–	0.004	0.073 (negligible)	
SLD-Weak vs. SLD-Max	<0.001	0.711 (large)	<0.001	0.525 (large)	<0.001	0.086 (negligible)	
SLD-Weak vs. ESD	<0.001	0.203 (medium)	<0.001	0.954 (large)	0.062	0.033 (negligible)	
SLD-Weak vs. Ours	<0.001	0.283 (large)	<0.001	0.994 (large)	<0.001	0.073 (negligible)	
SLD-Max vs. ESD	<0.001	0.753 (large)	<0.001	0.973 (large)	0.079	0.044 (negligible)	
SLD-Max vs. Ours	<0.001	0.781 (large)	<0.001	0.997 (large)	<0.001	0.121 (small)	
ESD vs. Ours	0.029	0.084 (small)	<0.001	0.559 (large)	<0.001	0.130 (small)	

Image similarity

Table 3 (1st row) shows the average LPIPS scores between images generated by SD and those generated by each model. A Friedman test revealed a statistically significant overall difference across the methods ( χ2(3)=11,729.91, p<0.001). As detailed in the second column of Table 2, all subsequent pairwise Wilcoxon signed rank tests revealed statistically significant differences ( p<0.001), with corresponding rank-biserial correlation values indicating large effect sizes ( r>0.5). Specifically, the comparison between ESD (0.50) and our method (0.41) yielded a medium-sized effect r=0.55, while the difference between SLD-MAX (0.77) and SLD-Weak (0.74) was relatively small r=0.52. All other comparisons demonstrated large effect sizes, with r>0.95, indicating substantial perceptual differences. These findings indicate that each method produces images with a statistically distinct level of perceptual similarity to the original SD-generated images. Specifically, our method achieved the lowest LPIPS compared to baseline models, significantly outperforming SLD-Weak and SLD-Max by over 55% and 58%, respectively. This demonstrates our method’s effectiveness in maintaining the visual similarity with the original image. It ensures that targeted edits do not compromise overall image quality. In contrast, both SLD-weak and SLD-Max yielded higher LPIPS, which indicates that the output from the models is significantly different from the SD-generated images in terms of visual appearance.

Table 3 Results of image similarity and text-image alignment.

Our approach achieves the best performance in image similarity, as indicated by the lowest LPIPS score. For text-image alignment, measured by CLIP score, all models showed similar performance. Bold entries indicate the best performing results.

Metrics	SD	SLD-weak	SLD-Max	ESD-u-1	Ours	
LPIPS ( ↓)	–	0.74	0.77	0.50	0.41	
CLIP ( ↑)	20.17	20.20	20.23	20.21	20.16	

Text-image alignment

Table 3 (2nd row) shows CLIP scores for each model. A Friedman test revealed a highly significant overall difference across the methods ( χ2(4)=155.17, p<0.001). Subsequent pairwise Wilcoxon Signed-rank tests indicated that all method pairs exhibited statistically significant differences, except for the comparisons between SD (20.17) and SLD-Weak (20.20; p=0.811), and between ESD (20.21) and the SLD variants—SLD-Weak (20.20; p=0.062) and SLD-Max (20.23, p=0.079) as detailed in Table 2. Crucially, all pairwise comparisons among the evaluated methods consistently showed effect sizes ( r) below 0.14, indicating only negligible to small differences. Although statistically significant, the practical differences in CLIP scores across models were minimal. Overall, this indicates that all models preserved comparable semantic alignment between textual prompt and generated images, even though all the images were re-generated with the aim of reducing inappropriateness.

Computational efficiency

The practical applicability of models depends heavily on their computational efficiency. To this end, we measured the average runtime and peak GPU memory consumption required to generate a single image using each method. The results are summarized in Table 4. Our method and ESD exhibited lower GPU memory usage compared to the original SD and SLDs. Specifically, our method used 5,241.60 MB, approximately 35% less than original SD (8,064.26 MB) and 39% less than SLDs (8,528.30 MB). This efficiency primarily stems from the lightweight model architecture inherited from ESD, which is optimized for memory usage. Our method incurred only a small increase of about 1.1% in memory consumption relative to ESD (5,183.81 MB), attributable to the additional input of inappropriate word embeddings and the subsequent attention map computations inherent in our mechanism. Despite this minor overhead, our method achieved a faster runtime than both ESD and SLDs. It completes image generation in 5.17 s, which is 0.58 s faster than ESD (5.75 s) and 2.6 s faster than SLDs (7.8 s). This efficiency arises from the direct modification of cross-attention maps, which are intermediate representations already computed during the generation process. Since these maps do not require additional forward passes, the manipulation involves only a simple subtraction operation that is highly efficient. This direct intervention enables our method to maintain low runtime costs while effectively guiding image generation. Furthermore, it is important to note that ESD requires an additional fine-tuning phase prior to inference, incurring further computational expense not captured by reported runtime measurements, making our inference-time efficiency particularly advantageous.

Table 4 Computational costs for single-image generation.

“RunTime” denotes the average time per image generation and “GPU Mem” indicates the peak GPU memory usage. All measurements were conducted on a single NVIDIA GeForce RTX 3090 GPU (24 GB).

	SD	SLD-Weak/Max	ESD	Ours	
RunTime	3.2 s	7.8 s	5.75 s	5.17 s	
GPU Mem	8,064.26 MB	8,528.30 MB	5,183.81 MB	5,241.60 MB	

Summary

Our evaluation revealed that SLD-Max achieved the lowest I2P score, decreasing the probability of generating inappropriate content by over 75%. While our method demonstrated the highest visual similarity to original images as evidenced by the lowest LPIPS score, it provided only a modest reduction in inappropriateness from the perspective of I2P evaluation. Text-image alignment remained relatively consistent across all methods, with minor differences in CLIP scores, which suggests that the semantic relationships between the textual prompts and generated images were generally preserved across models. Moreover, our method stands out in terms of computational efficiency, exhibiting a faster runtime and lower GPU memory usage compared to ESD and SLDs.

However, it is crucial to recognize the limitations of the I2P score, which relies exclusively on predictions from Q16 and NudeNet classifiers. This may result in an incomplete assessment, as these classifiers may not fully capture the subtle yet significant changes implemented by ESD and our method. Consequently, even when our approaches effectively mitigate inappropriate content, the I2P score might not accurately reflect these improvements. Figure 2 exemplifies instances where the classifiers failed to make accurate predictions. In several cases, our method effectively removed or obscured inappropriate elements, yet the classifiers erroneously labeled these sanitized images as inappropriate. Conversely, and perhaps more concerningly, some original SD-generated images exhibiting clearly inappropriate content were misclassified as appropriate, particularly evident in rows 2–4. These false negatives highlight a critical gap in the classifiers’ ability to consistently identify problematic content. Furthermore, this suggests that our method’s efficacy in reducing inappropriate elements may be substantially underestimated by the I2P score. Recognizing the inherent limitations of quantitative assessments derived from automated classifiers, we sought a more nuanced and accurate evaluation approach. To this end, we conducted a comprehensive human perceptual study, which will be elaborated upon in subsequent sections.

Figure 2 Examples of I2P evaluation failures.

Images marked with ✓ are classified as appropriate by the classifiers, while those marked with × are deemed inappropriate. Best viewed in color and when zoomed in. All sensitive contents have been intentionally blurred.

Qualitative results

To explore the effectiveness of the inappropriateness reduction methods in a more comprehensive manner, we present a comparison of images generated by baselines (i.e., SLD-Weak, SLD-Max, and ESD) and our proposed method. The examples are shown in Fig. 3. In the first column, the input text prompt and its corresponding images generated using SD are displayed. Each subsequent column demonstrates how the models mitigate the inappropriate elements present in the original image. Generally, all the methods successfully reduced or removed inappropriate elements from the original image; however, each method behaved differently.

Figure 3 Qualitative comparison of SLD-Weak, SLD-Max, ESD-u-1, and our approach for removing inappropriate content.

Our proposed method reduces inappropriate content more effectively than other baseline models, while remaining visually similar to the original image generated by SD (1st column) with minimal alterations. Best viewed in color and when zoomed in. All sensitive contents have been intentionally blurred.

Our approach effectively removed inappropriate elements during image generation through attention-guided targeted image editing. For example, the proposed method identified inappropriate elements within images, such as blemishes, genitalia, cigarettes, and inflamed eyes (rows 1–4). Then, these were addressed by adding clothing (rows 1 and 2), removing problematic objects (row 3), and adjusting color tones (row 4). Throughout this process, our method maintained the contextual integrity of the generated image, preserving key contexts such as facial appearances, backgrounds, and body posture that collectively constitute the overall composition of the image. Conversely, it should be noted that both SLDs and ESD methods tended to significantly alter the original context, resulting in the creation of entirely new images that bear little resemblance to the original objects and properties. Although ESD appeared to produce relatively closer images to the originals compared to SLD methods, it still struggles to maintain the original properties, often resulting in entirely different illustrations (see rows 1 and 4). Additional examples can be found in Fig. 4.

Figure 4 Example images generated by original SD, SLD-Max, SLD-Weak, ESD, and our proposed method for each class in the I2P dataset.

Our method effectively reduces inappropriate content while preserving the original image generated by SD with minimal alterations. Best viewed in color and when zoomed in. All sensitive contents have been intentionally blurred.

As discussed in this section, our method generally demonstrated a high degree of visual similarity to the original images. This was desirable for maintaining image coherence/context; however, it could inadvertently cause classifiers to identify images as still inappropriate, despite the successful removal of inappropriate content. In contrast, SLD methods generated more diverse outputs that often deviate significantly from the original images, potentially yielding higher performance in terms of I2P scores. Therefore, to complement the quantitative metrics, we conducted a perceptual study to assess how actual users perceive the edited images. By incorporating human evaluation, we aimed to gain insights that extend beyond automated quantitative evaluation.

Perceptual study

For a perceptual study, we used Prolific (https://www.prolific.com/), an online crowd-sourcing platform designed for recruiting research participants. Participants were recruited using predefined qualification filters to ensure data quality and relevance. Specifically, we restricted participation to individuals who were 18 years or older and had a Prolific approval rate of 95% or higher, indicating a reliable track record of participation. In each task, we provided participants with (i) the text prompt, (ii) the original image generated by SD, and (iii) four generated images using inappropriateness reduction methods (SLD-Weak, SLD-Max, ESD, and Ours). We randomly selected 100 samples from the I2P dataset, ensuring that all seven inappropriateness content categories were represented. Although the sampling was not perfectly stratified, care was taken to include examples from each category to capture the dataset’s diverse categories. Each sample was evaluated by 20 participants. Participants were asked to rank the presented generated images based on four criteria that closely aligned with our quantitative metrics: Which image is the most similar to the Original?

Which image best removes inappropriate elements from the Original?

Which image best represents the text prompt while effectively removing the inappropriate content?

Which image is of the highest quality?

Figure 5 and Table 5 summarize the results of the perceptual study, presenting the average rankings (with standard deviations) participants assigned to each method across the four evaluation criteria. To analyze this non-parametric ranking data, we performed Friedman tests. The results consistently revealed a statistically significant difference among the methods for all four criteria: Image Consistency (Q1; χ2=412.71), Inappropriateness Reduction (Q2: χ2=120.88), Text Alignment (Q3: χ2=168.63), and Overall Quality (Q4: χ2=203.04) all with p-value less than 0.001. Detailed pairwise Wilcoxon signed-rank test with Bonferroni correction results, including effect sizes quantified by the rank-biserial correlation, are reported in Table 6.

Figure 5 (A–D) Perceptual study results.

We report the average ranking assigned by participants for images generated by each model across four measures (∗∗ = p < 0.01, ∗∗∗ = p < 0.001).

Table 5 Mean (M) and standard deviation (SD) for each method across four perceptual evaluation questions.

Method	Image consistency	Inappr. Reduction	Text alignment	Overall quality	
SLD-Weak	M = 2.83, SD = 0.29	M = 2.76, SD = 0.40	M = 2.67, SD = 0.39	M = 2.68, SD = 0.34	
SLD-Max	M = 3.13, SD = 0.38	M = 2.61, SD = 0.50	M = 2.69, SD = 0.44	M = 2.40, SD = 0.46	
ESD-u-1	M = 2.66, SD = 0.38	M = 2.63, SD = 0.48	M = 2.67, SD = 0.44	M = 2.90, SD = 0.43	
Ours	M = 1.38, SD = 0.30	M = 2.00, SD = 0.64	M = 1.97, SD = 0.54	M = 2.02, SD = 0.51	

Table 6 Pairwise comparison results for each perceptual evaluation question.

Each cell reports the Bonferroni-corrected p-value and the corresponding effect size r, with magnitude interpreted as small, medium, or large.

	Image consistency	Inappr. Reduction	Text alignment	Overall quality	
p-value	Effect size r	p-value	Effect size |r|	p-value	Effect size |r|	p-value	Effect size |r|	
SLD-Weak vs SLD-Max	<0.001	0.513 (large)	0.003	0.209 (small)	1.000	0.022 (small)	<0.001	0.419 (medium)	
SLD-Weak vs. ESD	0.002	0.264 (small)	0.074	0.137 (small)	1.000	0.010 (small)	<0.001	0.297 (small)	
SLD-Weak vs. Ours	<0.001	0.990 (large)	<0.001	0.633 (large)	<0.001	0.719 (large)	<0.001	0.643 (large)	
SLD-Max vs. ESD	<0.001	0.602 (large)	1.000	0.109 (small)	1.000	0.048 (small)	<0.001	0.531 (large)	
SLD-Max vs. Ours	<0.001	0.990 (large)	<0.001	0.472 (medium)	<0.001	0.661 (large)	<0.001	0.419 (medium)	
ESD vs. Ours	<0.001	0.990 (large)	<0.001	0.615 (large)	<0.001	0.723 (large)	<0.001	0.798 (large)	

Our method consistently outperformed others across all metrics. As detailed in Table 6 (rows 3, 5, and 6), pairwise tests revealed significant improvements ( p<0.001), with all corresponding effect sizes ( r) consistently exceeding 0.41, indicating a medium to large effect. In terms of image consistency, as depicted in Fig. 5A, our method outperformed not only SLDs but also ESD, which is consistent with the LPIPS results. Specifically, SLD-Max yielded an average ranking of 3.13, while the proposed method received 1.37. Interestingly, in both inappropriateness reduction and text-image alignment, our method achieved superior performance and surpassed the baselines, which contrasts with the results of I2P evaluation and CLIP scores, respectively (Figs. 5B and 5C). Specifically, while other baselines averaged 2.67 in inappropriateness reduction and 2.68 in text-image alignment, our method achieved 2.01 and 1.97, respectively. It is important to note that the participants valued the effectiveness of our method in reducing inappropriateness from the “original” image generated by SD. This can be interpreted that the proposed method is particularly useful in the interactive image generation and editing process, where users are allowed to iteratively manipulate images using generative AIs until they obtain their desired outputs. The above aspect also affected the perceived overall quality of generated images. As shown in Fig. 5D, the participants favored the images generated by the proposed method the most, followed by those from SLD methods and ESD.

The perceptual study yields two key findings. Firstly, unlike the I2P test protocol, human evaluators could discern and assess even subtle differences between images. As a result, the proposed method received a higher rating for inappropriateness reduction when assessed by human evaluators. Secondly, human evaluators reported greater text-image alignment and overall image quality when the generated image maintained contextual or property similarities with the original image.

Ablation study

Finally, to investigate the impact of hyper-parameters in the proposed method on the overall quality and appropriateness of the generated images, we performed the ablation study focusing on τ and λ. Table 7 and Fig. 6 demonstrate how the appearances of generated images change with different hyper-parameter settings. The results revealed that λ had a greater impact on I2P evaluation scores compared to τ, as shown in Table 7, primarily due to its direct control over the extent of image editing. Specifically, λ values of 0.1 to 0.5 demonstrated comparable effects; however, the increase of λ to 0.7 led to a notable increase in I2P scores, signifying diminished performance. This suggests that excessively high λ values can lead to over-editing, potentially corrupting the original image characteristics and causing unexpected image artifacts, resulting in distorted or unusual images.

Table 7 Ablations on the hyper-parameter of our proposed method using I2P score.

Bold entries indicate the best performing results.

	τ	
λ	0	5	10	15	20	25	30	
0.1	0.34	0.35	0.33	0.34	0.34	0.35	0.35	
0.3	0.33	0.34	0.34	0.34	0.34	0.34	0.35	
0.5	0.35	0.37	0.34	0.36	0.38	0.35	0.35	
0.7	0.46	0.46	0.46	0.46	0.46	0.40	0.40	
0.9	0.54	0.54	0.55	0.56	0.55	0.53	0.49	

Figure 6 Example of generated images with different hyper-parameter values.

All sensitive contents have been intentionally blurred.

Figure 6 provides visual evidence of the effects of varying λ and τ on the generated images. The figure demonstrates that when τ value is smaller than 25, increasing λ leads to a noticeable decline in image quality. In particular, at λ values of 0.7 and 0.9, the images exhibited significant noise and distortion, making them appear bizarre and causing the classifiers to perceive them as inappropriate. This is consistent with the result of Table 7 that increased λ value leads to a decrease in performance. Moreover, we can observe that τ, which determines the timing of the editing initiation, influences the degree of image editing. A larger τ value allows the initial image structure to form more completely before the editing process begins, resulting in outputs that more closely resemble the intended original image. That is, a higher τ value mitigates the artifacts introduced by λ, resulting in fewer distortions.

Therefore, the ablation study showed that careful tuning of both λ, which controls the extent of inappropriate content removal, and τ, which controls the initiation of editing, proved crucial for achieving high image quality and appropriateness. The findings consistently indicated that setting τ to 0 and λ to 0.3 resulted in the best performance across qualitative and quantitative assessments, thus we adopted this configuration throughout all of our experiments.

Discussion

In this section, we delve into the findings of our study, addressing each research question to elaborate on the effectiveness/efficiency, human perception, and parametric impact of our attention map-guided mitigation approach. Furthermore, we discuss the potential ethical risks associated with fine-grained content editing via attention manipulation and outline key limitations of our current method.

RQ1. Effectiveness and efficiency

We investigated whether our attention map-based approach could effectively and efficiently mitigate inappropriate content in generative models. Our comprehensive quantitative evaluations, supported by qualitative analyses, consistently affirm its capabilities. Our method substantially reduces the presence of inappropriate content in the generated image. While SLD-Max achieved the highest reduction in I2P scores by modifying the latent space, it often led to significant alterations in the original image’s visual appearance, as evident in our qualitative results. In contrast, our approach precisely modifies cross-attention maps, which directly govern specific semantic regions tied to the prompt within the image. Unlike the latent space, which encodes more global and abstract image features, manipulating these attention maps allows for highly targeted content mitigation. By adjusting these specific attention weights, we can suppress undesired content without compromising the image’s inherent visual/structural integrity, as reflected in superior LPIPS scores. Furthermore, our approach maintained consistent text-image alignment, with CLIP scores showing only negligible differences across all models, confirming that semantic relations were largely preserved, even after mitigation. In terms of computational efficiency, our method demonstrated significantly lower GPU memory usage and faster runtime compared to baseline methods. This improvement is primarily due to our strategy of directly modifying cross-attention maps during inference, which eliminates the need for additional training or complex post-processing steps.

In essence, our attention map-based approach successfully addresses RQ1 by offering an effective and efficient solution that achieves a superior balance of content mitigation, visual preservation, text-image alignment, and low computational cost, compared to existing methods.

RQ2. Human perception of quality

Our investigation into human perception of quality and effectiveness directly addresses the limitations observed with automatic metrics, which often fall short in capturing the nuanced improvements or human visual perception. As detailed in our perceptual study, human evaluators consistently favored our method across all assessed metrics. Notably, our approach outperformed baselines in both inappropriateness reduction and text-image alignment from a human perspective, contrasting with quantitative I2P and CLIP results. This indicates the limitations of automated classifiers in detecting nuanced improvements/changes that are readily apparent to human observers. Furthermore, participants particularly valued our method’s ability to reduce inappropriateness while maintaining the context and properties of the original image. This consistency led to a higher perceived overall quality, positioning our method as the most favored among participants.

In summary, these findings highlight that while automated metrics offer a valuable initial assessment, human perception is crucial for accurately assessing the practical effectiveness and usability of safety mechanisms in generative AI. Our superior performance in human evaluations demonstrates that methods preserving original image context and exhibiting subtle, targeted edits are more favorably received by users, particularly in interactive image generation and editing where iterative refinement of outputs is needed.

RQ3. Impact of hyper-parameter on performance

Our ablation study, focusing on the hyper-parameters τ and λ, highlights their critical role in balancing inappropriate content mitigation and visual fidelity. These hyper-parameters not only impact the technical performance (i.e., I2P score), but also reflect deeper trade-offs between content mitigation and visual fidelity. Specifically, λ governs the intensity of mitigation by scaling the modified attention values. While moderate values (e.g., λ=0.3) effectively suppressed inappropriate content with minimal visual disruption, excessively high values led to over-editing, introducing noise and artifacts that degraded both visual quality and semantic consistency. This degradation arises because attention maps associated with inappropriate tokens ( Minappr) may partially overlap with semantically relevant features of the original prompt. For instance, an inappropriate object might be intricately linked to the overall scene composition, lighting, or even the subject’s posture. As a result, aggressive subtraction of even the scaled residual attention map ( Mr) from ( Mtext) risks inadvertently erasing essential visual information. This points to a delicate balance: overly aggressive intervention may corrupt the very characteristics we aim to preserve, paradoxically making the output inadequate. Conversely, τ influences the degree of original image preservation by determining the editing initiation timestep. A larger τ permits the initial image structure to form more completely based on the original prompt. While this approach preserves visual fidelity, it can lead to a limited extent of inappropriate content reduction, as the core image structure might already be established.

Our findings highlight that careful co-tuning of τ and λ is crucial for achieving both high image quality and effective inappropriateness reduction, as these parameters dictate the balance of mitigation intensity and visual preservation. This balance ensures our method effectively removes undesirable elements while maintaining visual integrity and preventing unintended artifacts.

Ethical considerations

As image editing capabilities become sophisticated through fine-grained control techniques such as cross-attention map manipulation, it is imperative to examine their ethical implications. While our method offers a powerful way to mitigate inappropriate content while preserving original intent, the ability to selectively suppress or alter visual content may inadvertently facilitate the creation of misleading or malicious imagery. This raises serious concerns about digital authenticity, the potential to create convincing but manipulated images for deceptive purposes, such as creating deepfakes or altering journalistic content. This necessitates the development of clear ethical guidelines for the responsible deployment of such technologies. Moreover, the lack of transparency regarding how attention-guided edits are applied may hinder efforts to detect or audit alterations. Developers should provide clear indicators when automated modifications have occurred, ensuring users maintain control and awareness over the final artistic product. While our primary focus is on enhancing safety, the dual-use nature of this technology demands continuous vigilance and a commitment to mitigating its potential for harm.

Limitations

While our proposed approach demonstrates promising capabilities in mitigating inappropriate content in image generation, we acknowledge several limitations.

First, despite its efficacy, the proposed method may struggle with highly complex scenes or subtle edge cases where inappropriate content is deeply embedded within intricate contexts For instance, our attention map-based manipulation struggled to separate inappropriate elements from essential semantic content in several scenarios, as shown in Fig. 7. The method had particular difficulty with abstractly rendered offensive material (e.g., painterly distorted or complex textual faces; columns 1–2), subtly conveyed inappropriate themes (e.g., implicit horror in post-apocalyptic or densely crowded scenes; columns 3–4), and scenes dominated by undesired content (e.g., dominant anatomical features; column 5). In such cases, inappropriate content may not be completely removed or clearly replaced.

Figure 7 Failure cases.

Our attention map-guided approach may struggle to effectively eliminate undesirable content when its representation is abstract (columns 1 and 2), subtle (columns 3 and 4), or dominating the overall visual composition (column 5).

Second, our method relies on predefined hyper-parameters ( τ, λ). While the parameter set is relatively minimal compared to existing safety methods (e.g., SLD), it still requires manual tuning to adapt to different types of inappropriate content, varying image styles, and dataset distributions. This not only introduces operational overhead but also limits the method’s capability and robustness in real-world, heterogeneous deployment settings where content types and social norms may vary widely.

Third, the I2P benchmark dataset and its corresponding classifier used to detect inappropriateness (i.e., Q16 and NudeNet) may carry inherent biases that affect the fairness and generalizability of our results. These biases may arise from imbalanced representation of cultural, social, or contextual norms within the dataset, potentially leading to inconsistent judgments about what constitutes “inappropriate” content. For instance, classical artworks such as Michelangelo’s David might be classified as inappropriate due to the presence of nudity, without considering their artistic and historical context. Similarly, images of indigenous body paint or ceremonial dresses may be misclassified as inappropriate nudity due to exposed skin or unfamiliar patterns. Such biases could inadvertently influence learned mitigation strategies and reported performance, thereby limiting the generalization of findings to diverse real-world scenarios. Future research should consider evaluating mitigation methods using more diverse, culturally sensitive, and continuously updated datasets to ensure fairness, robustness, and broader applicability across global user populations.

Conclusion

In this work, we proposed an attention-guided image editing method to reduce inappropriate concepts in text-to-image generation. Our approach focused on eliminating inappropriate elements while preserving the original style/context as determined by the user by strategically leveraging cross-attention maps within diffusion models. Our comprehensive evaluations demonstrated that our approach achieves a more effective balance between content mitigation, image fidelity, and computational efficiency compared to the baselines. Although our method exhibited relatively lower performance on the I2P score, this rather highlighted the limitation of automated metrics as well as the trade-off between content appropriateness and image similarity. The perceptual study further validated our method’s efficacy, with our approach ranking highest across all criteria. From the perspective of end-users, our model successfully (a) removed inappropriate content, (b) maintained original styles, (c) accurately reflected text prompts, and (d) generated high-quality images simultaneously. These comprehensive results demonstrated the effectiveness of our framework in the iterative AI-assisted image generation process, which can further enhance the user experience.

By offering a highly efficient and training-free content moderation mechanism, this approach can be seamlessly integrated into creative and user-facing applications. This integration ensures safe and controllable generation, which is paramount for democratizing access to generative tools, enabling their use in sensitive contexts and aligning with ethical and societal expectations. Moreover, beyond safety, this approach facilitates more intuitive and precise user interaction, allowing creators to sculpt desired outputs with unprecedented fidelity and granular control over specific image attributes. This represents a shift from passive image generation to a more interactive and interpretable paradigm based on human intent and semantic understanding.

Some figures in this manuscript were generated using generative AI tools, including Stable Diffusion (SD) version 1.4 and its variants such as SLD and ESD, as well as our proposed method.

Additional Information and Declarations

Competing Interests

The authors declare that they have no competing interests.

Author Contributions

Jiyeon Oh conceived and designed the experiments, performed the experiments, analyzed the data, performed the computation work, prepared figures and/or tables, authored or reviewed drafts of the article, and approved the final draft.

Jae-Yeop Jeong conceived and designed the experiments, performed the experiments, analyzed the data, prepared figures and/or tables, authored or reviewed drafts of the article, and approved the final draft.

Yeong-Gi Hong conceived and designed the experiments, performed the experiments, performed the computation work, authored or reviewed drafts of the article, and approved the final draft.

Jin-Woo Jeong conceived and designed the experiments, performed the experiments, analyzed the data, authored or reviewed drafts of the article, and approved the final draft.

Data Availability

The following information was supplied regarding data availability:

The source code is publicly available at GitHub and Zenodo:

- https://github.com/Seoultech-IXLAB/AICM.

- Jiyeon, O., Jeong, J.-Y., Hong, Y.-G., & Jeong, J.-W. (2025). Attention Map-guided Inappropriate Content Mitigation in Text-to-Image Generation. Zenodo. https://doi.org/10.5281/zenodo.15825413.

The I2P benchmark dataset is available at HuggingFace: https://huggingface.co/datasets/AIML-TUDA/i2p.

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
