# Peer review of "Mitigating inappropriate concepts in text-to-image generation with attention-guided Image editing"

_PeerJ Computer Science, doi:10.7717/peerj-cs.3170_

## Round 0.1 · original submission · Major Revisions

· Academic Editor

Major Revisions

While the study presents an interesting and relevant contribution to the field, some critical areas must be improved. Please revise the manuscript by incorporating the feedback below. Pay particular attention to Reviewer 1’s detailed suggestions for figures, methodology, and discussion expansion, as these are central to enhancing the clarity and impact of your work.

Address Reviewer 2’s observations through the annotated file to ensure all relevant points are considered.

Reviewer 1 ·

Basic reporting

The manuscript is well-structured and written in clear, professional English. The authors provide proper background and context in the introduction, thoroughly explaining the challenges of inappropriate content generation in text-to-image models. The literature is extensively referenced, covering relevant works in areas such as image generation safety, attention mechanisms, and image editing techniques.

The figures effectively illustrate the proposed method and results. Figure 1 demonstrates the attention map concept clearly, while Figures 2-6 provide good qualitative comparisons of the results. However, I suggest:

1. Figure 1's caption and labeling need more crucial details. The caption should explain the different color intensities in the attention maps, specify how these maps were generated, and clarify the relationship between Mtext, Minappr, and Mr.

2. The authors should improve Figure 5's readability by increasing the size of axis labels and adding clearer legends to help readers interpret the box plots more easily.

Experimental design

The research question is well-defined and meaningful. It addresses the important challenge of reducing inappropriate content while preserving image quality and original style. The authors clearly explain how their work fills a gap in existing approaches.

The methodology is rigorous and described with sufficient detail for replication. However, I suggest:

Your methodology section needs more detail in these areas:

1. Please provide specific details about the training/validation split used for the I2P dataset evaluation.

2. The process for selecting the 100 samples for the perceptual study should be better explained - was it truly random or stratified across different inappropriate content categories?

3. The criteria for participant selection in the perceptual study via Prolific should be detailed.

Validity of the findings

The experimental results are well-supported by both quantitative and qualitative evidence. The authors appropriately use multiple evaluation metrics (I2P score, LPIPS, CLIP score) and complement these with human perceptual studies. The conclusions are clearly stated and properly linked to the research questions.

However, I recommend addressing these limitations:

1. The authors should discuss potential biases in the I2P dataset and how these might affect the generalization of their results.

2. More statistical analysis of the perceptual study results would strengthen the findings - please include confidence intervals or statistical significance tests for the rankings.

3. The ablation study could include more parameter combinations to justify the chosen values better.

Additional comments

Some suggestions for improvement:

1. The discussion of limitations could be expanded, particularly regarding complex scenes and edge cases.

2. A more detailed analysis of computational costs compared to baseline methods would be valuable.

3. Consider adding a section on ethical considerations and potential misuse of the technology.

4. The conclusion could better highlight the broader implications for the field.

·

Basic reporting

The English is professional.
The literature is well-grounded.
Professional article structure is not followed.
The hypotheses are not clear.
Detailed proofs are not provided.

Experimental design

It aligns well with the aims and scope of the journal.
Research questions are not stated explicitly.
The gap is not explicitly mentioned.
The method section needs more details.

Validity of the findings

It seems that only a small portion of data are provided.
There is no clear statistical verifications of the results.
Discussion and conclusion are irrelevant. They are not linked to the research question and the literature.
The proposed claims of differences in the results section are not robust as they are not supported statistically.

Additional comments

I think the manuscript needs major revisions at this stage. However, I found the study very interesting and it intrigued me a lot. The findings can be very useful for readers if supported strongly. The proposed method can also be relevant to the literature.
My comments can be found in the revised manuscript.

---

## Round 0.2 · accepted · Accept

· Academic Editor

Accept

All reviewer comments have been adequately addressed by the authors, and I recommend the manuscript for acceptance.

·

Basic reporting

The article improved greatly after the revisions.

Experimental design

There is no problem

Validity of the findings

There is no problem

Additional comments

The revisions are thorough and acceptable.